# DEEP FUNCTION MACHINES: GENERALIZED NEURAL NETWORKS FOR TOPOLOGICAL LAYER EXPRESSION

## ABSTRACT

In this paper we propose a generalization of deep neural networks called deep function machines (DFMs). DFMs act on vector spaces of arbitrary (possibly infinite) dimension and we show that a family of DFMs are invariant to the dimension of input data; that is, the parameterization of the model does not directly hinge on the quality of the input (eg. high resolution images). Using this generalization we provide a new theory of universal approximation of bounded non-linear operators between function spaces. We then suggest that DFMs provide an expressive framework for designing new neural network layer types with topological considerations in mind. Finally, we introduce a novel architecture, RippLeNet, for resolution invariant computer vision, which empirically achieves state of the art invariance.

## 1 INTRODUCTION

In recent years, deep learning has radically transformed a majority of approaches to computer vision, reinforcement learning, and generative models [Schmidhuber (2015)]. Theoretically, we still lack a unified description of what computational mechanisms have made these deeper models more successful than their wider counterparts. Substantial analysis by Shalev-Shwartz et al. (2011), Raghu et al. (2016), Poole et al. (2016) and many others gives insight into how the *properties of neural architectures*, like depth and weight sharing, determine the expressivity of those architectures. However, less studied is the how the *properties of data*, such as sample statistics or geometric structure, determine the architectures which are most expressive on that data.

Surprisingly, the latter perspective leads to simple questions without answers rooted in theory. For example, what topological properties of images allow convolutional layers such expressivity and generalizeability thereon? Intuitively, spatial locality and translation invariance are sufficient justifications in practice, but is there a more general theory which suggests the optimality of convolutions? Furthermore, do there exist weight sharing schemes beyond convolutions and fully connected layers that give rise to provably more expressive models in practice? In this paper, we will more concretely study the data-architecture relationship and develop a theoretical framework for creating layers and architectures with provable properties subject to topological and geometric constraints imposed on the data.

**The Problem with Resolution.** To motivate a use for such a framework, we consider the problem of learning on high resolution data. Computationally, machine learning deals with discrete data, but frequently this data is sampled from a continuous process. For example, audio is inherently a continuous function $f : [0, t_{end}] \rightarrow \mathbb{R}$, but is sampled as a vector $v \in \mathbb{R}^{44,100 \times t}$. Even in vision, images are generally piecewise smooth functions $f : \mathbb{R}^2 \rightarrow \mathbb{R}^3$, but are sampled as tensors $v \in \mathbb{R}^{x \times y \times c}$. Performing tractible machine learning as the resolution of data of this type almost always requires some lossy preprocessing like PCA or Discrete Fourier Analysis [Burch (2001)]. Convolutional neural networks avoid dealing therein by intutively assuming a spacial locality on these vectors. However, one wonders what is lost through the use of various dimensionality reduction and weight sharing schemes[1].

---

[1] Note we do not claim that deep learning on high resolution data is currently intractible or ineffective. The problem of resolution is presented as an example in which topological constaints can be imposed on a type of data to yield new architecutres with desired, provable properties.

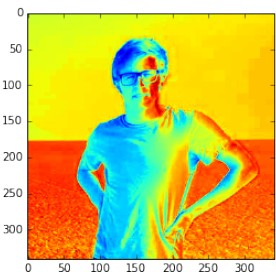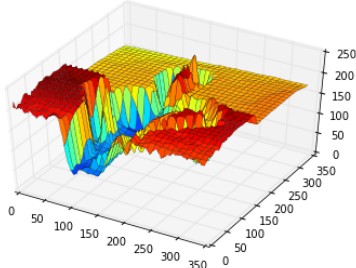

Figure 1: Left: A discrete vector $v \in \mathbb{R}^{l \times w}$ representation of an image. Right: The true continuous function $f : \mathbb{R}^2 \to \mathbb{R}$ from which it was sampled.

A key observation in discussing a large class of smooth functions is their simplicity. Although from a set theoretic perspective, the graph of a function consists of infiniteley many points, relatively complex algebras of functions can be described with symbolic simplicity. A great example are polynomials: the space of all square ($x^2$) monomials occupies a one-dimensional vector space, and one can generalize this phenomena beyond these basic families. Thus we will explore what results in embracing the assumption that a signal is really a sample from a continuous process, and utilize the analytic simplicity of certain smooth functions to derive new layer types.

**Our Contribution.** First, we extend neural networks to the infinite dimensional domain of continuous functions and define *deep function machines* (DFMs), a general family of function approximators which encapsolates this continuous relaxation and its discrete counterpart. Thereafer, we survey and refocus past analysis of neural networks with infinitely (and potentially uncountably) many nodes[2] with respect to the expresiveness the maps that they represent. We show that DFMs not only admit most other infinite dimensional neural network generalizations in the literature but also provide the necessary language to solve two long standing questions of universal approximation raised following Stinchcombe (1999). With the framework firmly established, we then return to our motivating goal of provable deep learning and show that DFMs naturally give rise to neural networks which are provably invariant to the resolution of the input, and indeed that DFMs can be used more generally to construct architectures (e.g. those with convolutions) with provable properties given topological assumptions. Finally we experimentally verify such constructions by introducing a new type of layer, WaveLayers, apart from convolutions.

## 2 BACKGROUND

In order to propose deep function machines we must establish what it means for a neural network act directly on continuous functions. Recall the standardMcCulloch & Pitts (1943) feed-forward neural network.

**Definition 2.1** (Discrete Neural Networks). *We say $\mathcal{N} : \mathbb{R}^n \to \mathbb{R}^m$ is a (discrete) feed-forward neural network iff for the following recurrence relation is defined for adjacent layers $\ell \to \ell'$,*

$$\mathcal{N} : y^{\ell'} = g\left(W_\ell^T y^\ell\right); \quad y_0 := x \tag{2.1}$$

*where $W_\ell$ is a weight tensor and $g$ is a non-polynomial activation function.*

Suppose that we wish to map one space of functions to another with a neural network. Consider the model of $\mathcal{N}$ as the number of neurons for every layer becomes uncountable. The index for each neuron then becomes real-valued, along with the weight and input vectors. The process is roughly depicted in Figure 2. The core idea behind the derivation is that as the number of nodes in the network becomes uncountable we need apply a normalizing term to the contribution of each node in the evaluation of the following layer so as to avoid saturation. Eventually this process resembles Lebesgue integration.

More formally, let $\mathcal{N}$ be an $L$ layer neural network as given in Definition 2.1. Without loss of generality we will examine the first layer, $\ell = 1$. Let us denote $\xi : X \subset \mathbb{R} \to \mathbb{R}$ as some

---

[2]See related work.

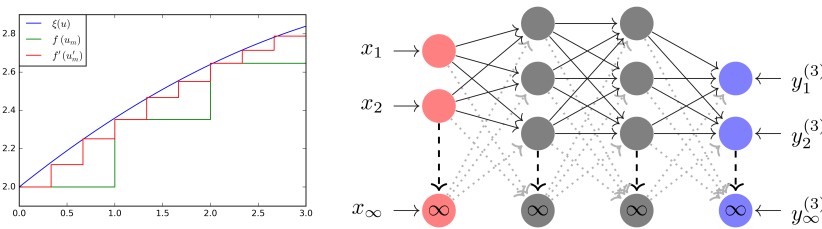

Figure 2: Left: Resolution refinement of an input signal by simple functions. Right: An illustration of the extension of neural networks to infinite dimensions. Note that $x \in \mathbb{R}^N$ is a sample of $f^{(N)}$, a simple function with $\|f^{(N)} - \xi\| \to 0$ as $N \to \infty$. Furthermore, the process is not actually countable, as depicted here.

arbitrary continuous input function for the neural network. Likewise consider a real-valued piecewise integrable *weight function*, $w_\ell : \mathbb{R}^2 \to \mathbb{R}$, for a layer $\ell$ which is composed of two indexing variables[3] $u, v \in E_\ell, E_{\ell'} \subset \mathbb{R}$. In this analysis we will restrict the indices to lie in compact sets $E_\ell, E_{\ell'}$.

If $f$ is a simple function then for some finite partition of $E_\ell$, say $u_0 < \cdots < u_n$, then $f = \sum_{m=1}^{n} \chi_{[u_{m-1}, u_m]} p_n$ where for all $u \in [u_{m-1}, u_m]$, $p_n \leq \xi(u)$. Visually this is a piecewise constant function underneath the graph of $\xi$. Suppose that some vector $x$ is sampled from $\xi$, then we can make $x$ a simple function by taking an arbitray parition of $E_\ell$ so that: when $u_0 < u < u_1$, $f(u) = x_0$, and when $u_1 < u < u_2$, $f(u) = x_1$, and so on. This simple function $f$ is essentially piecewise constant on intervals of uniform length so that on each interval it attains the value of the $n$th component, $x_n$. Finally if $w_v$ is some simple function approximating the $v$-th row of some weight matrix $W_\ell$ in the same fashion, then $w_v \cdot f$ is also a simple function. Therefore particular neural layer associated to $f$ (and thereby $x$) is

$$y^1 = g(W_\ell^T x) = g\left(\sum_{m=1}^{n} W_{mv}^\ell x_m \mu([u_{m-1}, u_m])\right) = g\left(\int_{E_\ell} w_v^\ell(u) f(u) \, d\mu(u)\right), \quad (2.2)$$

where $\mu$ is the Lebesgue measure on $\mathbb{R}$.

Now suppose that there is a refinement of $x$; that is, returning to our original problem, there is a higher resolution sample of $\xi$ say $f'$ (and thereby $x'$), so that it more closely approximates $\xi$. It then follows that the cooresponding refined partition, $u_0' < \cdots < u_k'$, (where $k > n$), occupies the same $E_\ell$ but individually, $\mu([u_{m-1}, u_m]) \leq \mu([u_{m-1}', u_m'])$. Therefore we weight the contribution of each $x_n'$ less than each $x_n$, in a measure theoretic sense.

Recalling the theory of simple functions without loss of generality assume $\xi, \omega(\cdot, \cdot) \geq 0$. Then we yield that if

$$F_v = \{(w_v, f) : E_\ell \to \mathbb{R} \mid f, w_v \text{ simple}, 0 \leq f \leq \xi, 0 \leq w_v \leq \omega_\ell(\cdot, v)\} \quad (2.3)$$

then it follows immediately that

$$\sup_{(f, w_v) \in F_v} \int_{E_\ell} w_v(u) f(u) \, d\mu(u) = \int_{E_\ell} \omega_\ell(u, v) \xi(u) \, d\mu(u). \quad (2.4)$$

Therefore we give the following definition for infinite dimensional neural networks.

**Definition 2.2** (Operator Neural Networks). *We call $\mathcal{O} : L^1(E_\ell) \to L^1(E_{\ell'})$ an operator neural network parameterized by $\omega_\ell$ if for two adjacent layers $\ell \to \ell'$*

$$\mathcal{O} : y^{\ell'}(v) = g\left(\int_{E_\ell} y^\ell(u) \omega_\ell(u, v) \, d\mu(u)\right); \quad y^0(v) = \xi(v). \quad (2.5)$$

*where $E_\ell, E_{\ell'}$ are locally compact Hausdorff mesure spaces and $u \in X, v \in Y$.*

---

[3]It is no loss of generality to extend the results in this work to weight kernels indexed by arbitrary $u, v \in \mathbb{R}^n$, but we ommit this treatment for ease of understanding.

## 3 DEEP FUNCTION MACHINES

With operator neural networks defined, we endeavour to define a topologically inspired framework for developing expressive layer types. A powerful language of abstraction for describing feed-forward (and potentially recurrent) neural network architectures is that of computational skeletons as introduced in Daniely et al. (2016). Recall the following definition.

**Definition 3.1.** *A computational skeleton $\mathcal{S}$ is a directed asyclic graph whose non-input nodes are labeled by activations.*

Daniely et al. (2016) provides an excellent account of how these graph structures abstract the many neural network architectures we see in practice. We will give these skeletons "flesh and skin" so to speak, and in doing so pursue a suitable generalization of neural networks which allows intermediate mappings between possibly infinite dimensional topological vector spaces. DFMs are that generalization.

**Definition 3.2** (Deep Function Machines). *A deep function machine $\mathcal{D}$ is a computational skeleton $\mathcal{S}$ indexed by $I$ with the following properties:*

- *Every vertex in $\mathcal{S}$ is a topological vector space $X_\ell$ where $\ell \in I$.*

- *If nodes $\ell \in A \subset I$ feed into $\ell'$ then the activation on $\ell'$ is denoted $y^\ell \in X_\ell$ and is defined as*

$$y^{\ell'} = g\left(\sum_{\ell \in A} T_\ell\left[y^\ell\right]\right) \tag{3.1}$$

*where $T_\ell : X_\ell \to X_{\ell'}$ is some affine form called the operation of node $\ell$.*

To see the expressive power of this generalization, we propose several operations $T_\ell$ that not only encapsulate ONNs and other abstractions on infinite dimensional neural networks, but also almost all feed-forward architectures used in practice.

### 3.1 GENERALIZED NEURAL LAYERS

Generalized neural layers are the basic units of the theory of deep function machines, and they can be used to construct architectures of neural networks with provable properties, such as the resolution invariance we seek. The most basic case is $X_\ell = \mathbb{R}^n$ and $X_{\ell'} = \mathbb{R}^m$, where we should expect a standard neural network. As either $X_\ell$ or $X_{\ell'}$ become infinite dimensional we hope to attain models of functional MLPs from Rossi et al. (2002) or infinite layer neural networks from Globerson & Livni (2016) with universal approximation properties.

**Definition 3.3** (Generalized Layer Operations). *We suggest several natural generalized layer families $T_\ell$ for DFMs as follows.*

- *$T_\ell$ is said to be $\mathfrak{o}$-operational if and only if $X_\ell$ and $X_{\ell'}$ are spaces of integrable functions over locally compact Hausdorff measure spaces, and*

$$T_\ell[y^\ell](v) = \mathfrak{o}(y^\ell)(v) = \int_{E_\ell} y^\ell(u)\omega_\ell(u,v)\,d\mu(u). \tag{3.2}$$

*For example[4], $X_\ell, X_{\ell'} = C(\mathbb{R})$, yields operator neural networks.*

- *$T_\ell$ is said to be $\mathfrak{n}$-discrete if and only if $X_\ell$ and $X_{\ell'}$ are finite dimensional vector spaces, and*

$$T_\ell[y^\ell] = \mathfrak{n}(y^\ell) = W_\ell^T y^\ell. \tag{3.3}$$

*For example, $X_\ell = \mathbb{R}^n$, $X_{\ell'} = \mathbb{R}^m$, yields standard feed-forward neural networks.*

- *$T_\ell$ is said to be $\mathfrak{f}$-functional if and only if $X_\ell$ is some space of integrable functions as mentioned previously and $X_{\ell'}$ is a finite dimensional vector space, and*

$$T_\ell[y^\ell] = \mathfrak{f}(y^\ell) = \int_{E_\ell} \omega_\ell(u)y^\ell(u)\,d\mu(u) \tag{3.4}$$

---

[4]Nothing precludes the definition from allowing multiple functions as input, the operation must just be carried on each coordinate function.

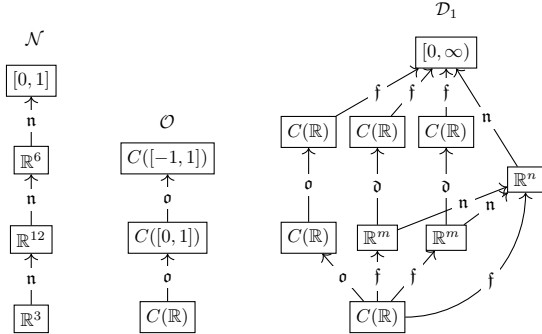

Figure 3: Examples of three different deep function machines with activations ommited and $T_\ell$ replaced with the actual type. Left: A standard feed forward binary classifier (without convolution), Middle: An operator neural network. Right: A complicated DFM with residues.

For example [5] $X_\ell = C(\mathbb{R})$, $X_{\ell'} = \mathbb{R}^n$, yields functional MLPs.

- $T_\ell$ is said to be $\mathfrak{d}$-defunctional if and only if $X_\ell$ is a are finite dimensional vector space and $X_{\ell'}$ is some space of integrable functions.

$$T_l[y^\ell](v) = \mathfrak{d}(y^\ell)(v) = \omega_\ell(v)^T y^\ell \tag{3.5}$$

For example, $X_\ell = \mathbb{R}^n$, $X_{\ell'} = C(\mathbb{R})$.

The naturality of the above layer operations come from their universality and generality.

## 3.2  RELATED WORK AND A UNIFIED VIEW OF INFINITE DIMENSIONAL NEURAL NETWORKS

Operator neural networks are just one of many instantiations of DFMs. Before we show universality results for deep function machines, it should be noted that there has been substantial effort in the literature to explore various embodiments of infinite dimensional neural networks. To the best of the authors' knowledge, DFMs provide a single unified view of every such proposed framework to date.

In particular, Neal (1996) proposed the first analysis of neural networks with countably infinite nodes, showing that as the number of nodes in discrete neural networks tends to infinity, they converge to a Gaussian process prior over functions. Later, Williams (1998) provided a deeper analysis of such a limit on neural networks. A great deal of effort was placed on analyzing covariance maps associated to the Guassian processes resultant from infinite neural networks with both sigmoidal and Gaussian activation functions. These results were based mostly in the framework of Bayesian learning, and led to a great deal of analyses of the relationship between non-parametric kernel methods and infinite networks, including Le Roux & Bengio (2007), Seeger (2004), Cho & Saul (2011), Hazan & Jaakkola (2015), and Globerson & Livni (2016).

Out of this initial work, Hazan & Jaakkola (2015) define hidden layer *infinite layer neural networks* with one or two layers which map a vector $x \in \mathbb{R}^n$ to a real value by considering infinitely many feature maps $\phi_w(x) = g(\langle w, x \rangle)$ where $w$ is an index variable in $\mathbb{R}^n$. Then for some weight function $u : \mathbb{R}^n \to \mathbb{R}$, the output of an infinite layer neural network is a real number $\int u(w) \phi_w(x) d\mu(w)$. This approach can be kernelized and therefore has resulted further theory by Globerson & Livni (2016) that aligns neural networks with Gaussian processes and kernel methods. Operatator neural networks differ significantly in that we let each $w$ be freely paramaterized by some function $\omega$ and require that $x$ be a continuous function on a locally compact Hausdorf space. Additionally no universal approximation theory is provided for infinite layer networks directly, but is cited as following from the work of Le Roux & Bengio (2007). As we will see, DFMs will not only encapsulate (and benefit from) these results, but also provide a general universal approximation theory therefor.

---

[5]Note that $y^\ell(u)$ is a scalar function and $\omega$ is a vector valuued function of dimension $dim(X_{\ell'})$. Additionally this definiton can easily be extended to function spaces on finite dimensional vectorspaces by using the Kronecker product.

Table 1: Unification of Infinite Dimensional Neural Network Theory

| Name | Form | DFM | Authors |
|------|------|-----|---------|
| Infinite NNs | $b + \sum_{j=1}^\infty v_j h(x; u_j)$ | $\mathcal{N}_\infty : \boxed{\mathbb{R}^n} \xrightarrow{\mathfrak{n}} \boxed{\oplus_{i=1}^\infty \mathbb{R}} \xrightarrow{\mathfrak{n}} \boxed{\mathbb{R}^m}$ | Neal (1996); Williams (1998) |
| Functional MLPs | $\sum_{i=1}^p \beta_i g\left(\int w_i \xi \, d\mu\right)$ | $\mathcal{F} : \boxed{L^1(\mathbb{R})} \xrightarrow{\mathfrak{f}} \boxed{\mathbb{R}^p} \xrightarrow{\mathfrak{n}} \boxed{\mathbb{R}}$ | Stinchcombe (1999); Rossi et al. (2002) |
| Continuous NNs | $\int \omega_1(u) g(x \cdot \omega_0(u)) \, du$ | $\mathcal{C} : \boxed{\mathbb{R}^n} \xrightarrow{\mathfrak{d}} \boxed{L^1([a,b])} \xrightarrow{\mathfrak{f}} \boxed{\mathbb{R}^m}$ | Le Roux & Bengio (2007) |
| Non-parametric Continuous NNs | $\int \omega_1(u) g(\langle x, u \rangle) \, du$ | $\mathcal{C}' : \boxed{\mathbb{R}^n} \xrightarrow{\mathfrak{d}} \boxed{L^1(\mathbb{R})} \xrightarrow{\mathfrak{f}} \boxed{\mathbb{R}^m}$ | Le Roux & Bengio (2007) |
| Infinite Layer NNs | same as non-parametric continuous NNs | same as non-parametric continuous NNs | Globerson & Livni (2016); Hazan & Jaakkola (2015) |

Another variant of infinite dimensional neural networks, which we hope to generalize, is the *functional multilayer perceptron* (Functional MLP). This body of work is not referenced in any of the aforementioned work on infinite layer neural networks, but it is clearly related. The fundamental idea is that given some $f \in V = C(X)$, where $X$ is a locally compact Hausdorff space, there exists a generalization of neural networks which approximates arbitrary continuous bounded functionals on $V$ (maps $f \mapsto a \in \mathbb{R}$). These functional MLPs take the form $\sum_{i=1}^p \beta_i g\left(\int \omega_i(x) f(x) \, d\mu(x)\right)$. The authors show the power of such an approximation using the functional analysis results of Stinchcombe (1999) and additionally provide statistical consistency results defining well defined optimal parameter estimation in the infinite dimensional case.

Stemming additionally from the initial work of Neal (1996), the final variant called *continuous neural networks* has two manifestations: the first of which is more closely related to functional perceptrons and the last of which is exactly the formulation of infintie layer NNs. Initially Le Roux & Bengio (2007) proposes an infinite dimensional neural network of the form $\int \omega_1(u) g(x \cdot \omega_0(u)) \, d\mu(u)$ and shows universal approximation in this regime. Overall this formulation mimics multiplication by some weighting vector as in infinite layer NNs, except in the continuous neural formulation $\omega_0$ can be parameterized by a set of weights. Thereafter, to prove connections between gaussian processes from a different vantage, they propose *non-parametric continuous neural networks*, $\int \omega_1(u) g(x \cdot u) \, d\mu(u)$, which are exactly infinite-layer neural networks.

In the view of deep function machines, the foregoing variants of infinite and semi-infinite dimensional neural networks are merely instantiations of different computational skeleton structures. A summary of the unified view is given in Table 1.

### 3.3 Approximation Theory of Deep Function Machines

In addition to unification, DFMs provide a powerful language for proving universal approximation theorems for neural networks of any depth and dimension[6]. The central theme of our approach is that *the approximation theories of any DFM can be factored through the standard approximation theories of discrete neural networks*. In the forthcoming section, this principle allows us to prove two approximation theories which have been open questions since Stinchcombe (1999).

The classic results of Cybenko (1989), yields the theory for $\mathfrak{n}$-discrete layers. For $\mathfrak{f}$-functional layers, the work of Stinchcombe (1999) proved in great generality that for certain topologies on $C(E_\ell)$, two layer functional MLPs universally approximate any continuous functional on $C(E_\ell)$. Following Stinchcombe (1999), Rossi et al. (2002) extended these results to the case wherein multiple

---

[6]By dimension, we mean both infinite and finite dimensional neural networks.

ɔ-operational layers prepended ʄ-functional layers. We will show in particular that ɔ-operational and similarly ᴆ-defunctional layers alone are dense in the much richer space of uniformly continuous bounded operators on function space. We give three results of increasing power, but decreasing transparency.

**Theorem 3.4** (Point Approximation). *Let $[a, b] \subset \mathbb{R}$ be a bounded interval and $g : \mathbb{R} \to B \subset \mathbb{R}$ be a continuous, bijective activation function. Then if $\xi : E_\ell \to \mathbb{R}$ and $f : E'_\ell \to B$ are $L^1(\mu)$ integrable functions there exists a unique class of ɔ-operational layers such that $g \circ \mathfrak{o}[\xi] = f$.*

*Proof.* We seek a class of weight kernels $\omega_\ell$ so that that $\mathfrak{o}[\xi] = f$. Let $\omega_\ell(u, v) = \left[ (g^{-1})' \circ (h(\Xi(u), v)) \right] h'(\Xi(u), v)$ where $\Xi(u)$ is the indefinite integral of $\xi$. Define $h$ so that it satisfies the following two equivalent equations $\mu$-a.e.

$$h(\Xi(b), v) - h(\Xi(a), v) = f(v)$$
$$\frac{\partial h(x, v)}{\partial x} \xi(u) \Big|_{x = \Xi(u), v = v, u \in \{a, b\}} = 0 \tag{3.6}$$

The proof is completed in the appendix. $\qquad\square$

The statement of Theorem 3.4 is not itself very powerful; we merely claim that ɔ-operational layers can at least map any one function to any other one function. However, the proof yields insight into what the weight kernels of ɔ-operational layers look like when the single condition $\xi \mapsto f$ is imposed. Therefrom, we conjecture but do not prove that a statstically optimal initialization for training ɔ-operational layers is given by satifying (3.6) when $\xi = \frac{1}{n} \sum_{n=1}^m \xi_n, f = \frac{1}{n} \sum_{n=1}^m f_n$, where the training set $\{(\xi_n, f_n)\}$ are drawn i.i.d from some distribution $D$.

**Theorem 3.5** (Nonlinear Operator Approximation). *Suppose that $E_1, E_2$ are bounded intervals in $\mathbb{R}$. For all $\kappa, \lambda$, if $K : Lip_\lambda(E_1) \to Lip_\kappa(E_3)$ is a uniformly continuous, nonlinear operator, then for every $\epsilon > 0$ there exists a deep function machine*

$$\mathcal{D} : \boxed{L^1(E_1)} \xrightarrow{\;\mathfrak{o}\;} \boxed{L^1(E_2)} \xrightarrow{\;\mathfrak{o}\;} \boxed{L^1(E_3)} \tag{3.7}$$

*such that $\left\| \mathcal{D}|_{Lip_\lambda} - K \right\| < \epsilon$.*

With two layer operator networks universal, it remains to consider ᴆ-deconvolutional layers.

**Theorem 3.6** (Nonlinear Basis Approximation). *Suppose $I, E_2, E_3$ are compact intervals, and let $C^\omega(X)$ denote the set of analytic functions on $X$. If $B : I^n \to C^\omega(E_3)$ is a continuous basis map to analytic functions then for every $\epsilon > 0$ there exists a deep function machine*

$$\mathcal{D} : \boxed{\mathbb{R}^n} \xrightarrow{\;\mathfrak{d}\;} \boxed{L^1(E_2)} \xrightarrow{\;\mathfrak{o}\;} \boxed{L^1(E_3)} \tag{3.8}$$

*such that $\|D|_{I^n} - B\| < \epsilon$ in the topology of uniform convergence.*

To the best of our knowledge, the above are the first approximation theorems for nonlinear operators and basis maps on function spaces for neural networks. The proofs in the appendix roughly involve a factorization of arbitrary DFMs through approximation theories for n-discrete layers.

Essentially, the factorization works as follows. In both of the foregoing theorems we want to roughly approximate some nonlinear map $K$ with a DFM $\mathcal{D}$. We therefore define an operator $|K|$, called an *affine projection*, that takes functions, converts them into piecewise constant approximations, applies $K$, and then again converts the result to piecewise constant approximations. Since there are a finite number, say $N$ and $M$, of pieces given in the input and the output of $|K|$ respectively, we can define an operator $\tilde{K} : \mathbb{R}^N \to \mathbb{R}^M$, called a *lattice map*, which in some sense reproduces $|K|$. We then show both Theorem 3.5 and Theorem 3.6 by approximating $\tilde{K}$ with a discrete neural network, $\mathcal{N}$, and chosing $\mathcal{D}$ to be such that its discreitzation is $\mathcal{N}$. Surprisingly, this principle holds for any DFM structure and a large class of different $K$ not just those which use piecewise constant approximations!

## 4 NEURAL TOPOLOGY FROM TOPOLOGY

As we have now shown, deep function machines can express arbitrarily powerful configurations of 'perceptron layer' mappings betwen different spaces. However, it is not yet theoretically clear how different configurations of the computaional skeleton and the particular spaces $X_\ell$ do or do not lead to a difference in expressiveness of DFMs. To answer questions of structure, we will return to the motivating example of high-resolution data, but now in the language of deep function machines.

### 4.1 RESOLUTION INVARIANT NEURAL NETWORKS

If an an input $(x_j) \in \mathbb{R}^N$ is sampled from an continuous function $\xi \in C(E_\ell)$, $\mathfrak{o}$-operational layers are a natural way of extending neural networks to deal directly with $\xi$. As before, it is useful to think of each $\mathfrak{o}$ as a continuous relaxation of a class of $\mathfrak{n}$, and from this perspective we can gain insight into the weight tensors of $\mathfrak{n}$-discrete layers as the resolution of $x$ increases.

**Theorem 4.1** (Invariance). *If $T_\ell$ is an $\mathfrak{o}$-operational layer with an integrable weight kernel $\omega(u, v)$ of $\mathcal{O}(1)$ parameters, then there is a unique fully connected $\mathfrak{n}$-discrete layer, $N_\ell$, with $\mathcal{O}(N)$ parameters so that $T_\ell[\xi](j) = N_\ell(x)_j$ for all $\xi, x$ as above.*

Theorem 4.1 is a statement of variance in parameterization; when the input is a sample of a smooth signal, fully connected $\mathfrak{n}$-discrete layers are naively overparameterized.

DFMs therefore yield a simple resolution invariance scheme for neural networks. Instead of placing arbitrary restrictions on $W_\ell$ like convolution or assuming that the gradient descent will implicitly find a smooth weight matrix or filter $W_\ell$ for $\mathfrak{n}$, we take $W_\ell$ to be the discretization of a smooth $\omega_\ell(u, v)$. An immediate advantage is that the weight surfaces, $\omega_\ell(u, v)$, of $\mathfrak{o}$-operational layers can be parameterized by dense families $f(u, v; w)$, whose parameters $w$ do not depend on the resolution of the input but on the complexity of the model being learnt.

**Resolution Invariance Schema**

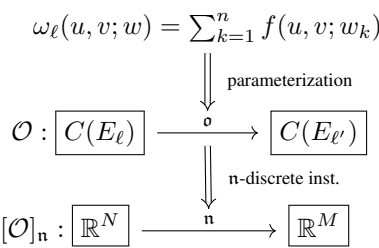

Figure 4: DFM construction of resolution invariant $\mathfrak{n}$-discrete layer.

### 4.2 TOPOLOGICALLY INSPIRED LAYER PARAMETERIZATIONS

Furthermore, we can now explore new parameterizations by constructing weight tensors and thereby neural network topologies which approximate the action of the operator neural networks which most expressively fit the topological properties of the data. Generally new restrictions on the weights of discrete neural networks might be achieved as follows:

1. Given that the input data $x$ is sampled from some $f \in \mathcal{F} \subset \{g : E_0 \to \mathbb{R}\}$, find a closed algebra of weight kernels so that $\omega_0 \in \mathcal{A}_0$ is minimally parameterized and $g \circ \mathfrak{o}[\mathcal{F}]$ is a sufficiently "rich" class of functions.

2. Repeat this process for each layer of a computational skeleton $\mathcal{S}$ and yield a DFM $\mathcal{O}$.

3. Instantiate a deep function machine $[\mathcal{O}]_\mathfrak{n}$ called *the $\mathfrak{n}$-discrete instatiation of $\mathcal{O}$* consisting of only $\mathfrak{n}$-discrete layers by discretizing each $\mathfrak{o}$-operational layer through the resolution invariance schema sample:

$$W_\ell = \left[ w^\ell \left( \frac{i}{e_{u_1}}, \frac{j}{e_{u_2}}, \cdots, \frac{k}{e_{v_1}}, \frac{t}{e_{v_2}}, \cdots \right) \right]_{ij\ldots kt\ldots} \tag{4.1}$$

where $e_\eta$ denotes the cardinality of the sample along the $\eta$-axis of $E_\ell = [0, 1]^{dim(E_\ell)}$. This process is depicted in Figure 4.

This perspective yields interpretations of existing layer types and the creation of new ones. For example, convolutional $\mathfrak{n}$-discrete layers provably approximate $\mathfrak{o}$-operational layers with weight kernels that are solutions to the ultrahyperbolic partial differential equation.

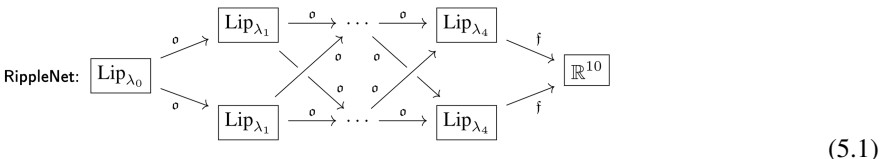

Figure 5: The WaveLayer architecture of RippLeNet for MNIST. Bracketed numbers denote number of wave coefficients. The images following each WaveLayer are the example activations of neurons after training given by feeding $'0'$ into RippLeNet.

**Theorem 4.2** (Convolutional Neural Networks). *Let $N_\ell$ be an $\mathfrak{n}$-discrete convolutional layer such that $\mathfrak{n}(x) = h \star x$ where $\star$ is the convolution operator and $h$ is a filter tensor. Then there is a $\mathfrak{o}$-operational layer, $O_\ell$ with $\omega_\ell(u_1, \ldots, u_n, v_1, \ldots, v_n)$ such that*

$$\sum_{k=1}^{n} \frac{\partial^2 \omega}{\partial^2 u_k} = c^2 \sum_{k=1}^{n} \frac{\partial^2 \omega}{\partial^2 v_k} \tag{4.2}$$

*and its $\mathfrak{n}$-discrete instatiation is $[O_\ell]_\mathfrak{n} = N_\ell$.*

Using Theorem 4.2 we therefore propose the following generalization of convolutional layers with weight kernels satisfying (4.2) whose $\mathfrak{n}$-discrete instantiation is resolution invariant.

**Definition 4.3** (WaveLayers). *We say that $T_\ell$ is a WaveLayer if it is the $\mathfrak{n}$-discrete instantiation (via the resolution invariance schema) of an $\mathfrak{o}$-operational layer with weight kernel of the form*

$$\omega_\ell(u, v) = s_0 + \sum_{i=1}^{b} s_i \cos(w_i^T(u, v) - p_i); \quad s_i, p_i \in \mathbb{R}, w_i \in \mathbb{R}^{2dim(E_\ell)}. \tag{4.3}$$

WaveLayers[7] are named as such, because the kernels $\omega_\ell$ are super position standing waves moving in directions encoded by $w_i$, offset in phase by $p_i$. Additionally any $\mathfrak{n}$-discrete convolutional layer can be expressed by WaveLayers, setting the direction $\theta_i$ of $w_i$ to $\theta_i = \pi/4$. In this case, instead of learning the values $h$ at each $j$, we learn $s_i, w_i, p_i$.

## 5 EXPERIMENTS

With theoretical guarantees given for DFMs, we propose a series of experiments to test the learnability, expressivity, and resolution invariance of WaveLayers.

**RippLeNet.** To find a baseline model, we performed a grid search on MNIST over various DFMs and hyperparameters, arriving at RippLeNet, an architecture similar to the classic LeNet-5 of LeCun et al. (1998) depicted in Figure 5. The model is the $\mathfrak{n}$-discrete instatiation of the following DFM

$$\text{(5.1)}$$

RippleNet: $\text{Lip}_{\lambda_0} \quad \text{Lip}_{\lambda_1} \xrightarrow{\mathfrak{o}} \cdots \xrightarrow{\mathfrak{o}} \text{Lip}_{\lambda_4} \quad \text{Lip}_{\lambda_1} \xrightarrow{\mathfrak{o}} \cdots \xrightarrow{\mathfrak{o}} \text{Lip}_{\lambda_4} \quad \mathbb{R}^{10}$$

and consiststs of 5 successive wave layers with $\tanh$ activations and no pooling. We found that using ReLu often resulted in a failure of learning to converge. Changes in the activation shapes (eg. 24x24x2 $\to$ 10x10x2) are achieved merely by specifying the sample partition of $E_\ell$ at each node of the DFM. The trainable parameters, in particular the magnitude of internal frequencies, were initialized at offsets proportional to the number of waves comprising each $\mathfrak{o}$-operational layer. Likewise, orientation of each wave was initialized uniformly on the unit spherical shell.

---

[7]*Note: WaveLayers are not not the same as Fourier networks or FC layers with $\cos$ activation functions.* It suffices to view wave layers as simply another way to reparameterize the weight matrix of $\mathfrak{n}$-discrete layers, and therefore the VC dimension of WaveLayers is less than that of normal fully connected layers. See the appendix.

**Model Expressive Parameter Reduction.** In Theorem 4.1, it was shown that DFMs in some sense parameterize the complexity of the model being learned, without constraints imposed by the form of data. RippLeNet benefits in that the sheer number of parameters (and therefore the variance) can be reduced until the model expresses the mapping to satisfactory error without concern for resolution variants.

We emprically verify this methodology by fixing the model architecture and increasing the number of waves per layer uniformly. This results in an exponential marginal utility on lowest error achieved after 80 epochs of MNIST with respect to the number of parameters in the model, shown in Figure 6. The slight outperformance of early LeNet architectures, suggest that future work in optimizing WaveLayers might be fruitful in achieving state of the art parameter reduction.

Figure 6: A plot of test error versus number of trainable parameters for RippLeNet in comparison with early LeNet architectures.

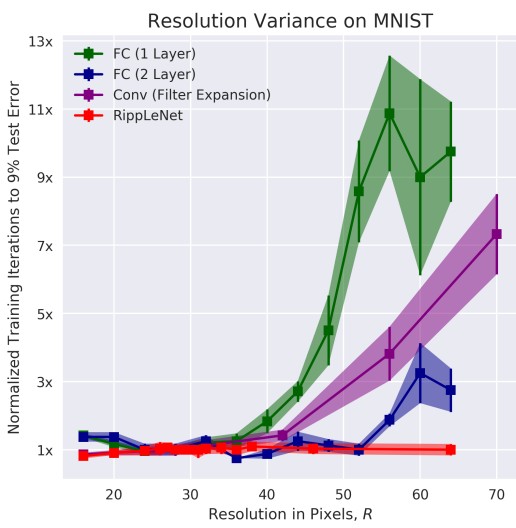

Figure 7: A plot of training time (normalized for each layer type with respect to the training time for $28 \times 28$ baseline) as the resolution of MNIST scales.

**Resolution Invariance.** True resolution invariance has the desirable property of consistency. Principly, consistency requires that regardless of the the resolution complexity of data, the training time, paramerization, and testing accuracy of a model do not vary.

We test consistency for RippLeNet by fixing all aspects of initialization save for the input resolution of images. For each training run, we rescale MNIST using both bicubic and nearest neighbor scaling to square resolutions of sidelength $R = \{16, \ldots, 36, 38, 46, 64, \ldots\}$. In conjuction we compare the resolution consistency of fully connected (FC) and convolutional architectures. For FC models, the number of free parameters on the first layer is increased out of necessity. Likewise, the size of the input filters for convolutional models is varied. As shown in Figure 7, WaveLayers remain invariant to resolution changes in both multirun variance and normalized convergence iterations, whereas both FC and convolutional layers exhibit an increase in both measurements with resolution.

## 6 CONCLUSION

In this paper we proposed deep function machines, a novel framework for topologically inspired layer parameterization. We showed that given topological assumptions, DFMs provide theoretical tools to yield provable properties in neural neural networks. We then used this framework to derive WaveLayers, a new type of provably resolution invariant layer for processing data sampled from continuous signals such as images and audio. The derivation of WaveLayers was additionally accompanied by the proposal of several layer operations for DFMs between infinite and/or finite dimensional vector spaces. We for the first time proved a theory of non-linear operator and functional basis approximation for neural networks of infinite dimensions closing two long standing questions since Stinchcombe (1999). We then utilized the expressive power of such DFMs to arrive at a novel architecture for resolution invariant image processing, RippLeNet.

**Future Work.** Although we've layed the ground work for exploration into the theory of deep function machines, there are still many open questions both theoretically and empirically. The drastic outperformance in resolution variance of RippLeNet in comparision to traditional layer types suggests that new layer types via DFMs with provable properties in mind should be further explored. Furthermore a deeper analysis of existing global network topologies using DFMs may be useful given their expressive power.

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

## 7 APPENDIX A: ADDITIONAL TOY DEMONSTRATION

To provide a direct comparison between convolutional, fully connected, and WaveLayer operations on data with semicontinuity assumptions, we conducted several toy demonstrations. We construct a dataset of functions (similar to a dataset of audio waveforms) whose input pairs are Gaussian bump functions, $B_\mu(u)$ centered at different points in the interval $\mu \in [0, 1]$. The coooresponding "labels" or target outputs are squres of gaussian bump functions plus a linear form whose slopes are given by the position of the center, that is $B_\mu(u)^2 + \mu(u - \mu)$. The desired map we wish to learn is then $T : B_\mu(u) \mapsto B_\mu(u)^2 + \mu \cdot (u - \mu)$. To construct the actual dataset $D$, for a random sample of centers $\mu$ we sample the input output pairs over 100 evenly spaced sub-intervals. The resultant dataset is a list of $N$ pairs of input/output vectors $D = \{(x_i, y_i)\}_{i=1}^{N}$ with $x, y \in \mathbb{R}^{100}$.

We then train three different two layer DFMs with n-discrete convolutional, fully-connected, and WaveLayers respectively. The following three figures show the outputs of all three layer types as training progresses. In the first three quadrants, the output of each layer type on a particular example datapoint is shown along with that example's input/target functions, $(x_i, y_i)$. The particular example shown is chosen randomly at the beginning of training. In the bottom right, the log training error over the whole dataset of each layer type is shown. A tick is the number of batches seen by the algorithm.

In initialization, the three layers exhibit predicted behavior despite artifacts towards the boundaries of the intervals. The convolutional layer, acts as a mullifer smoothing the input signal as its own kernel is Gaussian. The fully connected layer generates as predicted a normally distributed set of different output activations and does not regard the spatial locality (and thereby continuity) of the input. Finally the WaveLayer output exhibits behaviour predicted in Neal (1996), that is it limits towards a smoothed random walk over the input signal.

As training continues, both the convolutional and WaveLayer outputs preserve the continuity of the input signal, and approximate the smoothness of the output signal as induced by their own relation to ultrahyperbolic differential equations. Since the FC layer is not restricted to any given topology, although it approximates the desired output signal closely in the $L^2$ norm, it fails to achieve smoothness as this regularization is not explicity coded. It is important to note that in this example the WaveLayer output immediately surpasses the accuracy of the convolutional output because the convolutional output only has bias units accross entire channels, whereas the bias units of WaveLayers are themselves functions $\sum_{k=1}^{n} s_k cos(w_k \cdot v + p_k) + b_k$. Therefore WaveLayers can impose hetrogenously accross their output signals, where as convolutional require much deeper architectures to artifically generate such biases.

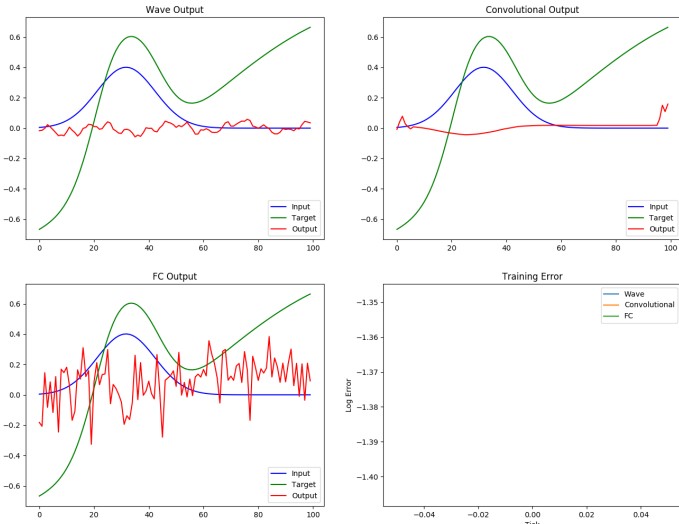

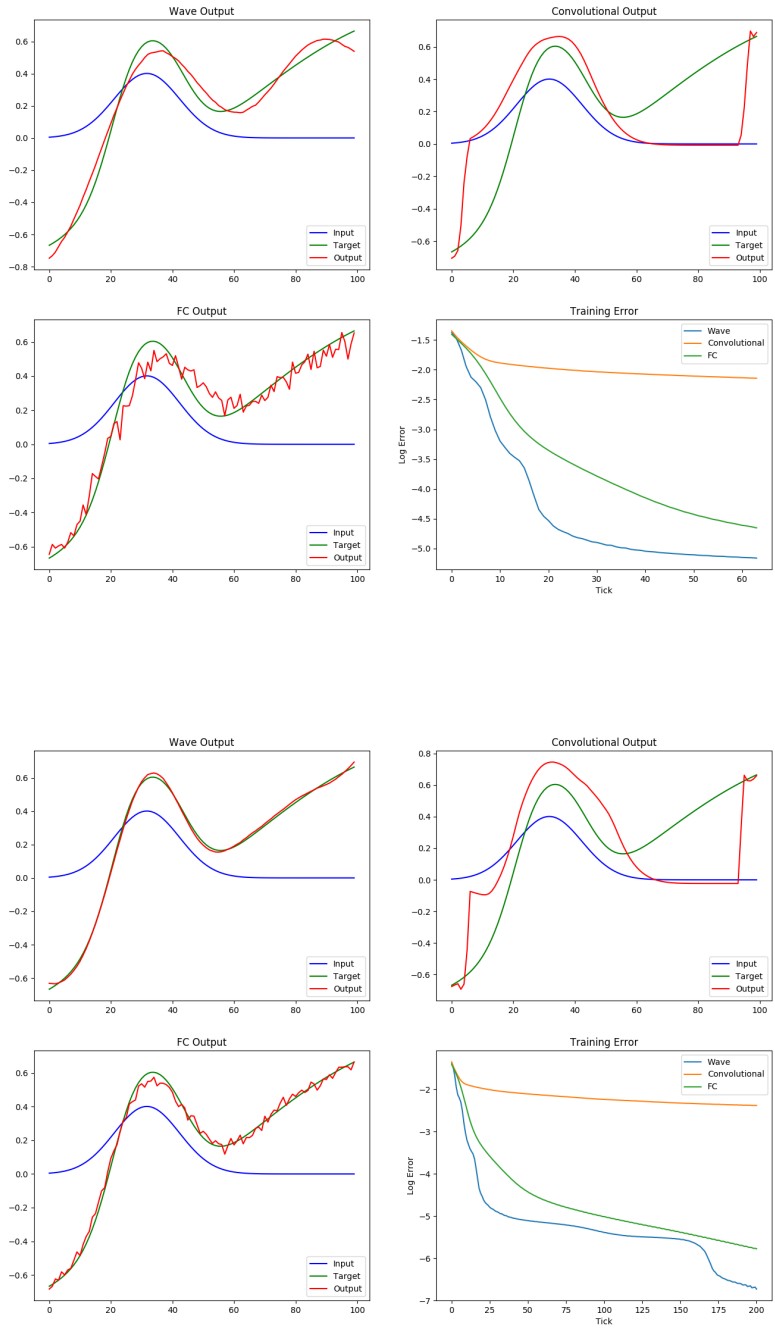

The results of this toy demonstration illustrate the intermediate flexibility of WaveLayers between purely fully connected and convolutional architectures. Satisfying the same differential equation (4.2), convolutional and WaveLayer architectures are regularized by spatial locality, but WaveLayers can in fact go beyond convolution layers and employ translational hetrogeneity. Although the purpose of this work is not to demonstrate the superiority of either convoluitional or WaveLayer architectures, it does open a new avenue of exploration in neural architecture design using DFMs to design layer types using topological constraints.

# 8 APPENDIX B: THEOREMS AND PROOFS

## 8.1 POINT APPROXIMATION

**Theorem 8.1.** *Let $[a, b] \subset \mathbb{R}$ be a bounded interval and $g : \mathbb{R} \to B \subset \mathbb{R}$ be a continuous, bijective activation function. Then if $\xi : E_\ell \to \mathbb{R}$ and $f : E'_\ell \to B$ are $L^1(\mu)$ integrable functions there exists a unique class of $\mathfrak{o}$-operational layers such that $g \circ \mathfrak{o}[\xi] = f$.*

*Proof.* We will give an exact formula for the weight function $\omega_\ell$ cooresponding to $\mathfrak{o}$ so that the formula is true. Recall that

$$y^{\ell'}(v) = g\left(\int_{E_\ell} \xi(u)\omega_\ell(u, v) \, d\mu(u)\right). \tag{8.1}$$

Then let $\omega_\ell(u, v) = \left[(g^{-1})' \circ (h(\Xi(u), v))\right] h'(\Xi(u), v)$ where $\Xi(u)$ is the indefinite integral of $\xi$ and $h : \mathbb{R} \times E_{\ell'} \to \mathbb{R}$ is some jointly and seperately integrable function. By the bijectivity of $g$ onto its codomain, $\omega_\ell$ exists. Now further specify $h$ so that, $h(\Xi(u), v)\Big|_{u \in E_\ell} = f(v)$. Then by the fundamental theorem of (Lebesgue) calculus and chain rule,

$$g(\mathfrak{o}[\xi](v)) = g\left(\int_{E_\ell} \left[(g^{-1})' \circ (h(\Xi(u), v))\right] h'(\Xi(u), v)\xi(u) \, d\mu(u)\right)$$

$$= g\left(g^{-1}\left(h(\Xi(u), v)\right)\right)\Big|_{u \in E_\ell} \tag{8.2}$$

$$= f(v)$$

A generalization of this theorem to $E_\ell \subset \mathbb{R}^n$ is given by Stokes theorem. $\qquad \square$

## 8.2 DENSITY IN LINEAR OPERATORS

**Theorem 8.2** (Approximation of Linear Operators). *Suppose $E_\ell$, $E_{\ell'}$ are $\sigma$-compact, locally compact, measurable, Hausdorff spaces. If $K : C(E_\ell) \to C(E'_\ell)$ is a bounded linear operator then there exists an $\mathfrak{o}$-operational layer such that for all $y^\ell \in C(E_\ell)$, $\mathfrak{o}[y^\ell] = K[y^\ell]$.*

*Proof.* Let $\zeta_t : C(E_{\ell'}) \to \mathbb{R}$ be a linear form which evaluates its arguments at $t \in E_{\ell'}$; that is, $\zeta_t(f) = f(t)$. Then because $\zeta_t$ is bounded on its domain, $\zeta_t \circ K = K^\star \zeta_t : C(E_\ell) \to \mathbb{R}$ is a bounded linear functional. Then from the Riesz Representation Theorem we have that there is a unique regular Borel measure $\mu_t$ on $E_\ell$ such that

$$\left(Ky^\ell\right)(t) = K^\star \zeta_t \left(y^\ell\right) = \int_{E_\ell} y^\ell(s) \, d\mu_t(s), \tag{8.3}$$

$$\|\mu_t\| = \|K^\star \zeta_t\|$$

We will show that $\kappa : t \mapsto K^\star \zeta_t$ is continuous. Take an open neighborhood of $K^\star \zeta_t$, say $V \subset [C(E_\ell)]^*$, in the weak* topology. Recall that the weak* topology endows $[C(E_\ell)]^*$ with smallest collection of open sets so that maps in $i(C(E_\ell)) \subset [C(E_\ell)]^{**}$ are continuous where $i : C(E_\ell) \to [C(E_\ell)]^{**}$ so that $i(f) = \hat{f} = \phi \mapsto \phi(f), \phi \in [C(E_\ell)]^*$. Then without loss of generality

$$V = \bigcap_{n=1}^m \hat{f}_{\alpha_n}^{-1}(U_{\alpha_n})$$

where $f_{\alpha_n} \in C(E_\ell)$ and $U_{\alpha_n}$ are open in $\mathbb{R}$. Now $\kappa^{-1}(V) = W$ is such that if $t \in W$ then $K^\star \zeta_t \in \bigcap_1^m \hat{f}_{\alpha_n}^{-1}(U_{\alpha_n})$. Therefore for all $f_{\alpha_n}$ then $K^*\zeta_t(f_{\alpha_n}) = \zeta_t(K[f_{\alpha_n}]) = K[f_{\alpha_n}](t) \in U_{\alpha_n}$.

We would like to show that there is an open neighborhood of $t$, say $D$, so that $D \subset W$ and $\kappa(Z) \subset V$. First since all the maps $K[f_{\alpha_n}] : E_{\ell'} \to \mathbb{R}$ are continuous let $D = \bigcap_1^m (K[f_{\alpha_n}])^{-1}(U_{\alpha_n}) \subset E_{\ell'}$. Then if $r \in D$, $\hat{f}_{\alpha_n}[K^\star \zeta_r] = K[f_{\alpha_n}](r) \in U_{\alpha_n}$ for all $1 \le n \le m$. Therefore $\kappa(r) \in V$ and so $\kappa(D) \subset V$.

As the norm $\| \cdot \|_*$ is continuous on $[C(E_\ell)]^*$, and $\kappa$ is continuous on $E_{\ell'}$, the map $t \mapsto \|\kappa(t)\|$ is continuous. In particular, for any compact subset of $E_{\ell'}$, say $F$, there is an $r \in F$ so that $\|\kappa(r)\|$ is maximal on $F$; that is, for all $t \in F$, $\|\mu_t\| \leq \|\mu_r\|$. Thus $\mu_t \ll \mu_r$.

Now we must construct a borel regular measure $\nu$ such that for all $t \in E_{\ell'}$, $\mu_t \ll \nu$. To do so, we will decompose $E_{\ell'}$ into a union of infinitely many compacta on which there is a maximal measure. Since $E_{\ell'}$ is a $\sigma$-compact locally compact Hausdorff space we can form a union $E_{\ell'} = \bigcup_1^\infty U_n$ of precompacts $U_n$ with the property that $U_n \subset U_{n+1}$. For each $n$ define $\nu_n$ so that $\chi_{U_n \setminus U_{n-1}} \mu_{t(n)}$ where $\mu_{t(n)}$ is the maximal measure on each compact $cl(U_n)$ as described in the above paragraph. Finally let $\nu = \sum_{n=1}^\infty \nu_n$. Clearly $\nu$ is a measure since every $\nu_n$ is mutually singular with $\nu_m$ when $n \neq m$. Additionally for all $t \in E_{\ell'}$, $\mu_t \ll \nu$.

Next by the Lebesgue-Radon-Nikodym theorem, for every $t$ there is an $L^1(\nu)$ function $K_t$ so that $d\mu_t(s) = K_t(s) \, d\nu(s)$. Thus it follows that

$$
\begin{aligned}
K\left[y^\ell\right](t) &= \int_{E_\ell} y^\ell(s) K_t(s) \, d\nu(s) \\
&= \int_{E_\ell} y^\ell(s) K(t,s) \, d\nu(s) = \mathfrak{o}[y^\ell](t).
\end{aligned}
\tag{8.4}
$$

By letting $\omega_\ell = K$ we then have $K = \mathfrak{o}$ up to a $\nu$-null set and this completes the proof. $\qquad \square$

## 8.3 Density in Non-Linear operators

**Theorem 8.3.** *Suppose that $E_1, E_2$ are bounded intervals in $\mathbb{R}$. If $K : Lip_\lambda(E_1) \to Lip_\kappa(E_3)$ is a uniformly continuous, nonlinear operator. Then for every $\epsilon > 0$ there exists a deep function machine*

$$
\mathcal{D} : \boxed{L^1(E_1)} \xrightarrow{\ \mathfrak{o}\ } \boxed{L^1(E_2)} \xrightarrow{\ \mathfrak{o}\ } \boxed{L^1(E_3)}
\tag{8.5}
$$

*such that $\left\| \mathcal{D}|_{Lip_\lambda} - K \right\| < \epsilon$.*

We will first introduce some defintions which quantize uniformly continuous operators on function space.

**Definition 8.4.** *Let $P = p_1 < \cdots < p_N$ be some partition of a compact interval $E$ with $N$ components. We call $\rho_P : Lip_*(E) \to \mathbb{R}^N$ and $\rho_P^* : \mathbb{R}^M \to Lip_*(E)$ affine projection maps if*

$$
\begin{aligned}
\rho_P(f) &= (f(p_i))_{i=1}^N \\
\rho_P^*(v) &= v \mapsto \sum_{i=0}^{N-1} \chi_{P_i}(x) \left[ \frac{(v_{i+1} - v_i)}{\mu(P_i)}(t - p_i) + v_i \right]
\end{aligned}
\tag{8.6}
$$

*where $\chi_{P_i}$ is the indicator function on $P_i = [p_i, p_{i+1})$ when $i < N$ and $P_{N-1} = [p_{N-1}, p_N]$.*

**Definition 8.5.** *Let $P, Q$ be partitions of $E_1, E_3$ of $N, M$ components respectively. If $K : Lip_\lambda(E_1) \to Lip_\kappa(E_3)$, its affine projection, $|K|$, and its lattice map, $\tilde{K}$, are defined so that the following diagram commutes,*

$$
\begin{array}{ccccc}
Lip_\lambda(E_1) & \xrightarrow{\ \rho_P\ } & \mathbb{R}^N & \xrightarrow{\ \rho_P^*\ } & Lip_\lambda(E_1) \\
\downarrow{\scriptstyle |K|} & & \downarrow{\scriptstyle \tilde{K}} & & \downarrow{\scriptstyle K} \\
Lip_\kappa(E_3) & \xleftarrow[\ \rho_Q^*\ ]{} & \mathbb{R}^M & \xleftarrow[\ \rho_Q\ ]{} & Lip_\kappa(E_3)
\end{array}
$$

**Lemma 8.6** (Strong Linear Approximation). *If $K : Lip_\lambda(E_1) \to Lip_\kappa(E_3)$ is a uniformly continuous, nonlinear operator, then for every $\epsilon > 0$ there exist partitions $P, Q$ of $E_1, E_3$ so that $\|K - |K|\| < \epsilon$.*

*Proof.* To show the lemma, we will chase the commutative diagram above by approximation.

For any $\delta > 0$, we claim that there exists a $P$ such that for any $f \in \text{Lip}_\lambda(E_1)$, the affine projection approximates $f$; that is, $\|f - \rho_P^* \circ \rho_P \circ f\|_{L_1(\mu)} < \delta$. To see this, take $P$ to be a uniform partition of

$E_1$ with $\Delta p := \mu(P_i) < \frac{\delta}{\mu(E_1)\lambda}$. Then

$$\int |f - \rho_P^* \circ \rho_P \circ f| \, d\mu \leq \sum_{i=1}^{N-1} \int_{P_i} \left| f(t) - \left[ \frac{(f(p_i) - f(p_i))}{\mu(P_i)} (t - p_i) + f(p_i) \right] \right| d\mu(t)$$

$$\leq \sum_{i=1}^{N-1} \int_{P_i} |f(t) - f(p_i) - \lambda(t - p_i)| \, d\mu(t)$$

$$\leq \sum_{i=1}^{N-1} \int_{P_i} 2\lambda |t - p_i| \, d\mu(t) \leq \lambda \Delta p^2 N < \delta.$$

Now by the absolute continuity of $K$, for every $\epsilon > 0$ there is a $\delta$ and therefore a partition $P$ of $E_1$ so that if $\|f - \rho_P^* \circ \rho_P \circ f\|_{L^1(\mu)} < \delta$ then $\|K[f] - K[\rho_P^* \circ \rho_P \circ f]\|_{L^1(\mu)} < \epsilon/2$. Finally let $Q$ be a uniform partition of $E_3$ so that for every $\phi \in \mathrm{Lip}_\kappa(E_3)$, $\|\phi - \rho_Q^* \circ \rho_Q \circ \phi\|_{L^1(\mu)} < \epsilon/2$. It follows that for every $f \in \mathrm{Lip}_\lambda(E_1)$

$$\|K[f] - |K|[f]\|_{L^1(\mu)} \leq \|K[f] - K \circ \rho_P^* \circ \rho_P[f]\| + \|K[f] - \rho_Q^* \circ \rho_Q \circ K[f]\|$$

$$< \frac{\epsilon}{2} + \frac{\epsilon}{2} = \epsilon.$$

Therefore the affine projection of $K$ approximates $K$. This completes the proof. □

With the lemma given we will approximate nonlinear operators through an approximation of the affine approximation using $\mathfrak{n}$-discrete DFMs.

*Proof of Theorem 3.5.* Let $\epsilon > 0$ be given. By Lemma 8.6 there exist partitions, $P, Q$, so that $\|K - |K|\| < \epsilon/2$. The cooresponding lattice map $\tilde{K} : \mathbb{R}^N \to \mathbb{R}^M$ is therefore continuous. Since $E_1$ is a compact interval, the image $\rho_P[\mathrm{Lip}_\lambda(E_1)]$ is compact and homeomorphic to the unit hypercube $[0,1]^N$. By the universal approximation theorem of Cybenko (1989), for every $\delta$, there exists a deep function machine

$$\mathcal{N} : \boxed{\mathbb{R}^N} \xrightarrow{\;\mathfrak{n}\;} \boxed{\mathbb{R}^J} \xrightarrow{\;\mathfrak{n}\;} \boxed{\mathbb{R}^M},$$

so that $\|\tilde{K} - \mathcal{N}\|_\infty < \delta$. Then, the continuity of the affine projection maps implies that there exist $\delta$ such that $\|\rho_Q^* \circ \mathcal{N} \circ \rho_P - |K|\| < \epsilon/2$. Therefore the induced operator on $\mathcal{N}$ represents $K$; that is, $\|\rho_Q^* \circ \mathcal{N} \circ \rho_P - K\| < \epsilon$.

Let $\mathcal{N}$ be parameterized by $W^1 \in \mathbb{R}^{N \times J}$ and $W^2 \in \mathbb{R}^{J \times M}$. Let $S$ be any uniform partition of an $I = [0,1]$ with $J$ components. Then parameterize a deep function machine $\mathcal{D}$ with weight kernels

$$\omega_1(u, v) = \sum_{i=1}^{N} \sum_{j=1}^{J} \chi_{S_j \times P_i}(u, v) W_{i,j}^1 \delta(u - p_i),$$

$$\omega_2(v, x) = \sum_{k=1}^{M-1} \chi_{Q_k}(x) \sum_{j=1}^{J} \left[ \frac{W_{j,k+1}^2 - W_{j,k}^2}{\mu(Q_k)} (x - q_k) + W_{j,k}^2 \right] \delta(v - s_j),$$

where $\delta$ is the dirac delta function. We claim that $\mathcal{D} = \rho_Q^* \circ \mathcal{N} \circ \rho_P$. Performing routine computations, for any $f \in Lip_\lambda(E_1)$,

$$\mathcal{D}[f] = T_2 \circ g \circ \left( \int_{E_1} f(u) \omega_1(u, v) \, d\mu(u) \right)$$

$$= T_2 \circ g \circ \left( \int_{E_1} \sum_{i=1}^{N} \sum_{j=1}^{J} \chi_{S_j \times P_i}(u, v) W_{i,j}^1 f(u) \delta(u - p_i) \, d\mu(u) \right)$$

$$= T_2 \circ g \circ \left( \sum_{j=1}^{J} \rho_P(f)^T W_j^1 \chi_{S_j}(v) \right) := T_2 \circ g \circ h(v)$$

Thus, $h$ is identical to the $j$th neuron of the first $\mathfrak{n}$-discrete layer in $\mathcal{N} \circ \rho_P$ when $v = s_j$. Turning to $T_2$ in $\mathcal{D}$, we get that

$$
\begin{aligned}
\mathcal{D}[f] &= \int_I \omega_2(v, x) g(h(v)) \, d\mu(v) \\
&= \sum_{k=1}^{M-1} \chi_{Q_k}(x) \sum_{j=1}^{J} \left[ \frac{W_{j,k+1}^2 - W_{j,k}^2}{\mu(Q_k)} (x - q_k) + W_{j,k}^2 \right] g(h(s_j)) \\
&= \sum_{k=1}^{M-1} \chi_{Q_k}(x) \cdot g(\rho_P(f)^T W^1)^T \left[ \frac{W_{k+1}^2 - W_k^2}{\mu(Q_k)} (x - q_k) + W_k^2 \right] \\
&= \rho_Q^*(g(\rho_P(f)^T W^1)^T W^2) = \rho_Q^* \circ \mathcal{N} \circ \rho_P[f].
\end{aligned}
$$

Therefore $\|\mathcal{D}|_{\mathrm{Lip}_\lambda} - K\| < \epsilon$ and this completes the proof.

$\square$

We will now prove a similar theorem for $\mathfrak{d}$-defunctional layers.

**Theorem 8.7** (Nonlinear Basis Approximation). *Suppose $I, E_2, E_3$ are compact intervals, and let $C^\omega(X)$ denote the set of analytic functions on $X$. If $B : I^n \to C^\omega(E_3)$ is a continuous basis map to analytic functions then for every $\epsilon > 0$ there exists a deep function machine*

$$
\mathcal{D} : \boxed{\mathbb{R}^n} \xrightarrow{\ \mathfrak{d}\ } \boxed{L^1(E_2)} \xrightarrow{\ \mathfrak{o}\ } \boxed{L^1(E_3)} \tag{8.7}
$$

*such that $\|D|_{I^n} - B\| < \epsilon$ in the topolopgy of uniform convergence.*

*Proof.* Recall that the set of polynomials on $E_3$, $\mathcal{P}$, are a basis for the vector space $C^\omega(E_3)$. Therefore the map $B$ has a decomposition through nmaots $\kappa, \Delta$ so that the following diagram commutes

$$
\begin{array}{ccc}
& & \ell^1(\mathbb{R}) \\
& {\scriptstyle \Delta} \nearrow & \downarrow {\scriptstyle \kappa} \\
\mathbb{R}^n & \xrightarrow{\ B\ } & C^\omega(E_3)
\end{array}
$$

and $\kappa : (a_i)_{i=1}^\infty \mapsto \sum a_n g_n$ where $g_n$ is the mononomial of degree $n$. The existence of $\Delta$ can be verified through a composition of the direct product of basis projections in $C^\omega(E_3)$ and $B$.

For each $m \in \mathbb{N}$ the projection image in the $m$th coordinate, $\pi_m[\Delta[\mathbb{R}^n]] = \mathbb{R}$ and so again the maos factor into a countable collection of maps $(\Delta_i : \mathbb{R}^n \to \mathbb{R})_{i=1}^\infty$ so that $\prod_{i=1}^\infty \Delta_i = \Delta$. We will approximate $B$ by approximations of $\kappa \circ \Delta$ via increasing products of $\Delta_i$.

Define the aforementioned increasing product map $\Delta^{(N)}$ as

$$
\Delta^{(N)} = \prod_{i=1}^N \Delta_i \times \prod_{N+1}^\infty c_0
$$

where $c_0$ is the constant map. Now with $\epsilon > 0$ given, we wish to show that there exists an $N$ so that $\|\kappa \circ \Delta^{(N)} - B\| < \epsilon$ in the topology of uniform convergence.

To see this let $\mathcal{P}_N \subset \mathcal{P}$ denote the set of polynomials of degree at most $n$. Next we define a open 'mullification' of $\mathcal{P}_N$. In particular let

$$
O\mathcal{P}_N(\epsilon) = \{ f \in C^\omega(\epsilon) \mid \ |f - g| < \epsilon, g \in \mathcal{P}_N \}.
$$

It is clear that $O\mathcal{P}_{N_1}(\epsilon) \subset O\mathcal{P}_{N_2}(\epsilon)$ when $N_1 \le N_2$ and furthermore by the density of $\mathcal{P}$ in $C^\omega(E_3)$ we have that $\{O\mathcal{P}_i(\epsilon)\}_{i=1}^\infty$ is an open cover of $B[I^n] \subset C^\omega(E_3)$. Since $I^n$ is compact $B[I^n]$ is a compact subset of $C^\omega(E_3)$ and thus there is a finite index set $I' = \{N_1, \dots N_k\}$ so that $\bigcup_{t \in I'} O\mathcal{P}_t \supset B[I^n]$. If $N = \max I'$ then $O\mathcal{P}_N(\epsilon) \supset B[I^n]$. Therefore for every $x \in I^n$ we have that $\|\kappa \circ \Delta^{(N)} - B\| < \epsilon$ since $\kappa \circ \Delta^{(N)}$ is a polynomial of degree at most $N$.

Now we will filter the maps $\psi_N := \pi_{1...N} \circ \Delta^{(N)}$ where $\psi_N : \mathbb{R}^n \to \mathbb{R}^N$ through the universal approximation theory of standard discrete neural networks. Let $\mathcal{N}$ be a two $\mathfrak{n}$-discrete layer DFM so that $\|\mathcal{N} - \psi_N\| < \epsilon$. For convienience let $\mathcal{N} := \mathfrak{n}_2 \circ g\mathfrak{n}_1$ Then we can instantiate $\mathcal{N}$ as the DFM in (8.7) using the same method as in the proof of the nonlinear operator approximation theory above.

For the $\mathfrak{d}$-defunctional layer let $W_{jk}^1$ be the weight tensor of $\mathfrak{n}_1$ in $\mathcal{N}$ so that $\mathfrak{n}_1(x)_j^1 = \sum_k W_{jk}^1 x_k$. Then let the weight kernel for $\mathfrak{d}$ be $\omega_1^k(v) = \rho^*(W_{\cdot k}^1)$. and then $\mathfrak{d}[x]|_{v=j} = \mathfrak{n}_1(x)_j^1$. We will ommit the design of weight kernels for the $\mathfrak{o}$-operational layer, but this is not difficult to establish. All together we now have that via the approximation of $\mathcal{N}$ and equivalence of $\mathcal{N}$ and its instantiation in the statement of the theorem,

$$\|\kappa \circ \rho \circ \mathfrak{o} \circ g \circ \mathfrak{d} - \kappa \circ \iota_N \circ \psi_N\| < \epsilon.$$

Finally we need deal with the basis map $\kappa$. On the compact set $\Delta^{(N)}[I^n] = \iota_N \circ \psi_N[I^n]$, $\kappa$ is a bounded linear operator and its composition, $\kappa \circ \rho \circ \mathfrak{o}$ is also a bounded linear operator. Therefore by the bounded linear approximation theorem of $\mathfrak{o}$-operational layers, there is a $\|\mathfrak{o}' - \kappa \circ \rho \circ \mathfrak{o}\| < \epsilon$. Appending such $\mathfrak{o}'$ to $\mathfrak{d}$ as above we achieve the approximation bound of the theorem. This completes the proof. □

## 8.4 RESOLUTION INVAIRANCE

**Theorem 8.8.** *If $T_\ell$ is an $\mathfrak{o}$-operational layer with an integrable weight kernel $\omega(u,v)$ of $\mathcal{O}(1)$ parameters, then there is a unique $\mathfrak{n}$-discrete layer with with $\mathcal{O}(N)$ parameters so that $\mathfrak{o}[\xi](j) = \mathfrak{n}[x]_j$ for all indices $j$ and for all $\xi, x$ as above.*

*Proof.* Given some $\mathfrak{o}$, we will give a direct computation of the corresponding weight matrix of $\mathfrak{n}$. It follows that

$$\begin{aligned}
\mathfrak{o}[\xi](v) &= \int_{E_\ell} \xi(u)\, \omega_\ell(u,v)\, d\mu(u) \\
&= \sum_n^{N-1} \int_n^{n+1} \left((x_{n+1} - x_n)(u-n) + x_n\right) \omega_\ell(u,v)\, d\mu(u) \\
&= \sum_n^{N-1} (x_{n+1} - x_n) \int_n^{n+1} (u-n)\omega_\ell(u,v)\, d\mu(u) + x_n \int_n^{n+1} \omega_\ell(u,v)\, d\mu(u)
\end{aligned} \tag{8.8}$$

Now, let $V_n(v) = \int_n^{n+1}(u-n)\omega_\ell(u,v)\, d\mu(u)$ and $Q_n(v) = \int_n^{n+1} \omega_\ell(u,v)\, d\mu(u)$; We can now easily simplify (8.8) using the telescoping trick of summation.

$$\mathfrak{o}(\xi)[v] = x_N V_{N-1}(v) + \sum_{n=2}^{N-1} x_n \left(Q_n(v) - V_n(v) + V_{n-1}(v)\right) + x_1 \left(Q_1(v) - V_1(v)\right) \tag{8.9}$$

Given indices in $j \in \{1, \cdots, M\}$, let $W \in \mathbb{R}^{N \times M}$ so that $W_{n,j} = (Q_n(j) - V_n(j) + V_{n-1}(j))$, $W_{N,j} = V_{N-1}(j)$, and $W_{1,j} = Q_1(j) - V_1(j)$. It follows that if $W$ parameterizes some $\mathfrak{n}$, then $\mathfrak{n}[x]_j = \mathfrak{o}[\xi](j)$ for every $f$ sampled/approximated by $x$ and $\xi$. Furthermore, $dim(W) \in O(N)$, and $\mathfrak{n}$ is unique up to $L^1(\mu)$ equivalence. □

## 8.5 CONVOLUTIONAL NEURAL NETWORKS AND THE ULTRAHYPERBOLIC DIFFERENTIAL EQUATION

*Proof.* A general solution to (4.3) is of the form $\omega(u,v) = F(u - cv) + G(u + cv)$ where $F, G$ are second-differentiable. Essentially the shape of $\omega$ stays constant in $u$, but the position of $\omega$ varies in $v$. For every $h$ there exists a continuous $F$ so that $F(j) = h_j$, $G = 0$. Let $\omega(u,v) = F(u - cv) + G(u + cv)$. Therefore applying Theorem 4.1, to $\mathfrak{o}$ parameterized by $\omega$, we yield a weight matrix $W$ so that

$$[\mathfrak{o}[\xi](j) = \int_{E_0} \xi(u)\left(F(u - cj) + 0\right) d\mu(u) = (Wx)_j = (h \star x)_j = \mathfrak{n}[x]_j. \tag{8.10}$$

This completes the proof.

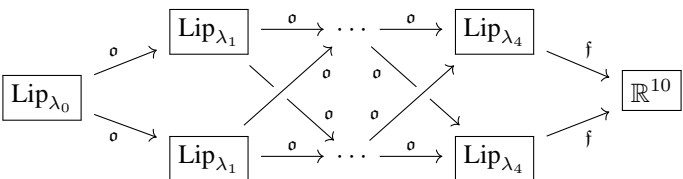

$\square$

# 9   APPENDIX C: VC DIMENSION OF DISCRETIZED OPERATOR NEURAL NETWORKS.

In order to calculate the VC dimension of DFMs contianing only discretized ø-operational layers, denoted $\mathfrak{D}$, we have $\mathfrak{D} \subset \mathfrak{N}$, where $\mathfrak{N}$ is the family of all DFMs with $\mathfrak{n}$-discrete skeletons whose per-node dimensionality is exactly that of the discretization $\mathfrak{D}$. Thus the VC dimension of $\mathfrak{F}$ can be bounded by that of $\mathfrak{N}$, however more fine tuned estimate is both possible and essential.

Suppose that in designing some deep architecture, one wishes to keep VC dimension low, whilst increasing per-node activation dimensionality. In practice optimization in higher dimensions is easier when a low dimensional parameterization is embedded therein. For example, hyperdimensional computing, sparse coding, and convolutional neural networks naturally neccessitate high dimensional hidden spaces but benefit from regularized capacity. Since the dimensionality of the discretization $\mathfrak{D}$ does not depend on the original dimensionality of the space, then the capacity of $\mathfrak{D}$ depends directly on the "complexity" of the family of weight surfaces there endowed. It would therefore be convenient to answer the following question formally.

**The VC Problem**. Let $\mathcal{W}_\ell \subset L^1(\mathbb{R}^2, \mu)$ be some family of weight surfaces. Then induce $\mathfrak{D}_{\mathcal{W}}$, a family of discretized ø-operational layers with $\mathfrak{D}_{\mathcal{W}} := \{[\mathfrak{o}_W]_{\mathfrak{n}}\}_{W \in \mathcal{W}}$ where $[\cdot]_{\mathfrak{n}}$ denotes the discretization. *What is $VCDim(\mathfrak{D}_{\mathcal{W}})$?*

Although in this work we do not directly attack this problem, a solution leads to another dimension of layer and architecture design beyond topological constraints. In practice, one would be able to choose which set of $\mathcal{W}_\ell$ to give a satisfactory generalizability condition on their learning problem.

# 10   APPENDIX D: ANALYTICAL DERIVATION OF CONTINUOUS ERROR BACKPROPAGATION FOR SEPERABLE WEIGHT KERNELS

With these theoretical guarantees given for DFMs, the implementation of the feedforward and error backpropagation algorithms in this context is an essential next step. We will consider operator neural networks with polynomial kernels. As aforementioned, in the case where a DFM has nodes with non-seperable kernels, we cannot give the guarntees we do in the following section. Therefore, a standard auto-differentiation set-up will suffice for DFMs with for example wave layers.

Feedforward propagation is straight forward, and relies on memoizing operators by using the separability of weight polynomials. Essentially, integration need only occur once to yield coefficients on power functions. See Algorithm 1.

### 10.0.1   FEED-FORWARD PROPAGATION

We will say that a function $f : \mathbb{R}^2 \to \mathbb{R}$ is numerically integrable if it can be seperated into $f(x, y) = g(x)h(y)$.

**Theorem 10.1.** *If $\mathcal{O}$ is a operator neural network with $L$ consecutive layers, then given any $\ell$ such that $0 \leq \ell < L$, $y^\ell$ is numerically integrable, and if $\xi$ is any continuous and Riemann integrable input function, then $\mathcal{O}[\xi]$ is numerically integrable.*

---

**Algorithm 1** Feedforward Propagation on $\mathcal{F}$

---

   **Input:** input function $\xi$
   **for** $l \in \{0, \ldots, L-1\}$ **do**
      **for** $t \in Z_X^\ell$ **do**
         Calculate $I_t^\ell = \int_{E_\ell} y^\ell(j_\ell) j_\ell^t \, dj_\ell$.
      **end for**
      **for** $s \in Z_Y^\ell$ **do**
         Calculate $C_s^\ell = \sum_a^{Z_X^\ell} k_{a,s} I_a^\ell$.
      **end for**
      Memoize $y^\ell(j) = g\left( \sum_b^{Z_Y^\ell} j^b C_b^\ell \right)$.
   **end for**
   The output is given by $\mathcal{O}[\xi] = y^L$.

---

*Proof.* Consider the first layer. We can write the sigmoidal output of the $(\ell)^{\text{th}}$ layer as a function of the previous layer; that is,

$$y^\ell = g\left( \int_{E_\ell} w^\ell(j_\ell, j_\ell) y^\ell(j_l) \, dj_l \right). \tag{10.1}$$

Clearly this composition can be expanded using the polynomial definition of the weight surface. Hence

$$
\begin{aligned}
y^\ell &= g\left( \int_{E_\ell} y^\ell(j_\ell) \sum_{x_{2\ell}}^{Z_Y^\ell} \sum_{x_{2l}}^{Z_X^\ell} k_{x_{2l}, x_{2\ell}} j_\ell^{x_{2l}} j_\ell^{x_{2\ell}} \, dj_\ell \right) \\
&= g\left( \sum_{x_{2\ell}}^{Z_Y^\ell} j^{x_{2\ell}} \sum_{x_{2l}}^{Z_X^\ell} k_{x_{2l}, x_{2\ell}} \int_{E_\ell} y^\ell(j_\ell) j_\ell^{x_{2l}} \, dj_\ell \right),
\end{aligned}
\tag{10.2}
$$

and therefore $y^\ell$ is numerically integrable. For the purpose of constructing an algorithm, let $I_{x_{2\ell}}^\ell$ be the evaluation of the integral in the above definition for any given $x_{2\ell}$

It is important to note that the previous proof requires that $y^\ell$ be Riemann integrable. Hence, with $\xi$ satisfying those conditions it follows that every $y^\ell$ is integrable inductively. That is, because $y^0$ is integrable it follows that by the numerical integrability of all $l$, $\mathcal{O}[\xi] = y^L$ is numerically integrable. This completes the proof. $\qquad\square$

Using the logic of the previous proof, it follows that the development of some inductive algorithm is possible.

### 10.0.2 CONTINUOUS ERROR BACKPROPAGATION

As is common with many non-convex problems with discretized neural networks, a stochastic gradient descent method will be developed using a continuous analogue to error backpropagation. We define the loss function as follows.

**Definition 10.2.** *For a operator neural network $\mathcal{O}$ and a dataset $\{(\gamma_n(j), \delta_n(j))\}$ we say that the error for a given $n$ is defined by*

$$E = \frac{1}{2} \int_{E_L} \left( \mathcal{O}(\gamma_n) - \delta_n \right)^2 \, dj_L \tag{10.3}$$

This error definition follows from $\mathcal{N}$ as the typical error function for $\mathcal{N}$ is just the square norm of the difference of the desired and predicted output vectors. In this case we use the $L^2$ norm on $C(E_L)$ in the same fashion.

We first propose the following lemma as to aid in our derivation of a computationally suitable error backpropagation algorithm.

**Lemma 10.3.** *Given some layer, $l > 0$, in $\mathcal{O}$, functions of the form $\Psi^\ell = g'\left( \Sigma_l y^\ell \right)$ are numerically integrable.*

*Proof.* If

$$\Psi^\ell = g' \left( \int_{E_{(\ell-1)}} y^{(\ell-1)} w^{(\ell-1)} \, dj_\ell \right) \tag{10.4}$$

then

$$\Psi^\ell = g' \left( \sum_b^{Z_Y^{(\ell-1)}} j_l^b \sum_a^{Z_X^{(\ell-1)}} k_{a,b}^{(\ell-1)} \int_{E_{(\ell-1)}} y^{(\ell-1)} j_\ell^a \, dj_{l-2} \right) \tag{10.5}$$

hence $\Psi$ can be numerically integrated and thereby evaluated. $\qquad\square$

The ability to simplify the derivative of the output of each layer greatly reduces the computational time of the error backpropagation. It becomes a function defined on the interval of integration of the next iterated integral.

**Theorem 10.4.** *The gradient, $\nabla E(\gamma, \delta)$, for the error function* (10.3) *on some $\mathcal{O}$ can be evaluated numerically.*

*Proof.* Recall that $E$ over $\mathcal{O}$ is composed of $k_{x,y}^\ell$ for $x \in Z_X^\ell, y \in Z_Y^\ell$, and $0 \le l \le L$. If we show that $\frac{\partial E}{\partial k_{x,y}^\ell}$ can be numerically evaluated for arbitrary, $l, x, y$, then every component of $\nabla E$ is numerically evaluable and hence $\nabla E$ can be numerically evaluated. Given some arbitrary $l$ in $\mathcal{O}$, let $n = \ell$. We will examine the particular partial derivative for the case that $n = 1$, and then for arbitrary $n$, induct over each iterated integral.

Consider the following expansion for $n = 1$,

$$\frac{\partial E}{\partial k_{x,y}^{L-n}} = \frac{\partial}{\partial k_{x,y}^{L-1}} \frac{1}{2} \int_{E_\ell} [\mathcal{O}(\gamma) - \delta]^2 \, dj_L$$

$$= \int_{E_\ell} [\mathcal{O}(\gamma) - \delta] \Psi^L \int_{E_{(\ell-1)}} j_{L-1}^x j_L^y y^{L-1} dj_{L-1} \, dj_L$$

$$= \int_{E_\ell} [\mathcal{O}(\gamma) - \delta] \Psi^L j_L^y \int_{E_{(\ell-1)}} j_{L-1}^x y^{L-1} dj_{L-1} \, dj_L \tag{10.6}$$

Since the second integral in (10.6) is exactly $I_x^{L-1}$ from (**??**), it follows that

$$\frac{\partial E}{\partial k_{x,y}^{(n)}} = I_x^{L-1} \int_{E_\ell} [\mathcal{O}(\gamma) - \delta] \Psi^L j_L^y \, dj_L \tag{10.7}$$

and clearly for the case of $n = 1$, the theorem holds.

Now we will show that this is all the case for larger $n$. It will become clear why we have chosen to include $n = 1$ in the proof upon expansion of the pratial derivative in these higher order cases.

Let us expand the gradient for $n \in \{2, \dots, L\}$.

$$\frac{\partial E}{\partial k_{x,y}^{L-n}} = \int_{E_L} [\mathcal{O}(\gamma) - \delta] \Psi^L \underbrace{\int_{E_{L-1}} w^{L-1} \Psi^{L-1} \int \cdots \int_{E_{L-n+1)}} w^{L-n+1)} \Psi^{L-n+1)}}_{n-1 \text{ iterated integrals}} \tag{10.8}$$

$$\int_{E_{L-n}} y^{L-n} j_{L-n}^a j_{L-n+1}^b \, dj_{L-n} \cdots dj_L$$

As aforementioned, proving the $n = 1$ case is required because for $n = 1$, (10.8) has a section of $n - 1 = 0$ iterated integrals which cannot be possible for the proceeding logic.

We now use the order invariance properly of iterated integrals (that is, $\int_A \int_B f(x,y) \, dxdy = \int_B \int_A f(x,y) \, dydx$) and reverse the order of integration of (10.8).

In order to reverse the order of integration we must ensure each iterated integral has an integrand which contains variables which are guaranteed integration over some region. To examine this, we propose the following recurrence relation for the gradient.

Let $\{B_s\}$ be defined along $L - n \leq s \leq L$, as follows

$$B_L = \int_{E_L} [\mathcal{O}(\gamma) - \delta] \, \Psi^L B_{L-1} \, dj_L,$$

$$B_s = \int_{E_\ell} \Psi^\ell \sum_a^{Z_X^\ell} \sum_b^{Z_Y^\ell} j_\ell^a j_\ell^b B_\ell \, dj_\ell, \tag{10.9}$$

$$B_{L-n} = \int_{E_{L-n}} j_{L-n}^x j_{L-n+1}^y \, dj_{L-n}$$

such that $\frac{\partial E}{\partial k_{x,y}^\ell} = B_L$. If we wish to reverse the order of integration, we must find a reoccurrence relation on a sequence, $\{Ⴈ_s\}$ such that $\frac{\partial E}{\partial k_{x,y}^{L-n}} = Ⴈ_{L-n} = B_L$. Consider the gradual reversal of (10.8).

Just as important as Clearly,

$$\frac{\partial E}{\partial k_{x,y}^\ell} = \int_{E_{L-n}} y^{L-n} j_{L-n}^x \int_{E_L} [\mathcal{O}(\gamma) - \delta] \Psi^L \int_{E_{L-1}} w^{L-1} \Psi^{L-1}$$

$$\int \cdots \int_{E_{L-n+1)}} j_{L-n+1}^y w^{L-n+1)} \Psi^{L-n+1)} \, dj_{L-n+1} \ldots dj_L dj_{L-n} \tag{10.10}$$

is the first order reversal of (10.8). We now show the second order case with first weight function expanded.

$$\frac{\partial E}{\partial k_{x,y}^\ell} = \int_{E_{L-n}} y^{L-n} j_{L-n}^x \int_{E_{L-n+1)}} \sum_b^{Z_Y} \sum_a^{Z_X} k_{a,b} j_{L-n+1}^{a+y} \Psi^{L-n+1)} \int_{E_L} [\mathcal{O}(\gamma) - \delta] \Psi^L$$

$$\int \cdots \int_{E_{L-n+1)}} j_{L-n+2}^b w^{(L-n+2)} \Psi^{(L-n+2)} \, dj_{L-n+1} \ldots dj_L dj_{L-n}. \tag{10.11}$$

Repeated iteration of the method seen in (10.10) and (10.11), where the inner most integral is moved to the outside of the $(L - s)^{\text{th}}$ iterated integral, with $s$ is the iteration, yields the following full reversal of (10.8). For notational simplicity recall that $l = L - n$, then

$$\frac{\partial E}{\partial k_{x,y}^\ell} = \int_{E_\ell} y^\ell j_l^x \int_{E_\ell} \sum_a^{Z_X^\ell} j_\ell^{a+y} \Psi^\ell \int_{E_{\ell+2}} \sum_b^{Z_Y^\ell} \sum_c^{Z_X^{\ell+2}} k_{a,b}^\ell j_{l+2}^{b+c} \Psi^{\ell+2}$$

$$\int_{E_{\ell+3}} \sum_d^{Z_Y^{\ell+2}} \sum_e^{Z_X^{\ell+3}} k_{c,d}^{\ell+2} j_{l+3}^{d+e} \Psi^{\ell+3} \int \cdots \int_{E_L} \sum_q^{Z_Y^{L-1}} k_{p,q}^{L-1} j_L^q \, [\mathcal{O}(\gamma) - \delta] \Psi^L \tag{10.12}$$

$$dj_L \ldots dj_{L-n}.$$

Observing the reversal in (10.12), we yield the following recurrence relation for $\{Ⴈ_s\}$. Bare in mind, $l = L - n$, $x$ and $y$ still correspond with $\frac{\partial E}{\partial k_{x,y}^\ell}$, and the following relation uses its definition on $s$ for cases not otherwise defined.

$$Ⴈ_{L,t} = \int_{E_L} \sum_b^{Z_Y^{L-1}} k_{t,b}^{L-1} j_L^b \, [\mathcal{O}(\gamma) - \delta] \, \Psi^L \, dj_L.$$

$$Ⴈ_{s,t} = \int_{E_{(s)}} \sum_b^{Z_Y^{(s-1)}} \sum_a^{Z_X^{(s)}} k_{t,b}^{(s-1)} j_s^{a+b} \Psi^{(s)} Ⴈ_{s+1,a} \, dj_s. \tag{10.13}$$

$$Ⴈ_\ell = \int_{E_\ell} \sum_a^{Z_X^\ell} j_\ell^{a+y} \Psi^\ell Ⴈ_{l+2,a} \, dj_\ell.$$

$$\frac{\partial E}{\partial k_{x,y}^\ell} = Ⴈ_l = \int_{E_\ell} j_l^x y^\ell Ⴈ_\ell \, dj_\ell.$$

---

**Algorithm 2** Error Backpropagation

---

**Input:** input $\gamma$, desired $\delta$, learning rate $\alpha$, time $t$.
**for** $\ell \in \{0, \ldots, L\}$ **do**
    Calculate $\Psi^\ell = g'\left(\int_{E_{(\ell-1)}} y^{(\ell-1)} w^{(\ell-1)}\, dj_\ell\right)$
**end for**
For every $t$, compute $\mathit{B}_{L,t}$ from from (10.13).
Update the output coefficient matrix $k_{x,y}^{L-1} - I_x^{L-1} \int_{E_L} [\mathcal{F}(\gamma) - \delta]\, \Psi^L j_L^y\, dj_L \to k_{x,y}^{L-1}$.
**for** $l = L - 2$ **to** $0$ **do**
    If it is null, compute and memoize $\mathit{B}_{l+2,t}$ from (10.13).
    Compute but do not store $\mathit{B}_\ell \in \mathbb{R}$.
    Compute $\frac{\partial E}{\partial k_{x,y}^\ell} = \mathit{B}_l$ from from (10.13).
    Update the weights on layer $l$: $k_{x,y}^\ell(t) \to k_{x,y}^\ell$
**end for**

---

Note that $\mathit{B}_{L-n} = B_L$ by this logic.

With (10.13), we need only show that $\mathit{B}_{L-n}$ is integrable. Hence we induct on $L - n \leq s \leq L$ over $\{\mathit{B}_s\}$ under the proposition that $\mathit{B}_s$ is not only numerically integrable but also constant.

Consider the base case $s = L$. For every $t$, because every function in the integrand of $\mathit{B}_L$ in (10.13) is composed of $j_L$, functions of the form $\mathit{B}_L$ must be numerically integrable and clearly, $\mathit{B}_L \in \mathbb{R}$.

Now suppose that $\mathit{B}_{s+1,t}$ is numerically integrable and constant. Then, trivially, $\mathit{B}_{s,u}$ is also numerically integrable by the contents of the integrand in (10.13) and $\mathit{B}_{s,u} \in \mathbb{R}$. Hence, the proposition that $s + 1$ implies $s$ holds for $\ell < s < L$.

Lastly we must show that both $\mathit{B}_\ell$ and $\mathit{B}_l$ are numerically integrable. By induction $\mathit{B}_{l+2}$ must be numerically integrable. Hence by the contents of its integrand $\mathit{B}_\ell$ must also be numerically integrable and real. As a result, $\mathit{B}_l = \frac{\partial E}{\partial k_{x,y}^\ell}$ is real and numerically integrable.

Since we have shown that $\frac{\partial E}{\partial k_{x,y}^\ell}$ is numerically integrable, $\nabla E$ must therefore be numerically evaluable as aforementioned. This completes the proof. $\square$

