# OpenReview forum: "Deep Function Machines: Generalized Neural Networks for Topological Layer Expression"
_ICLR.cc/2018/Conference — Reject_

### Official Review · AnonReviewer2 · 2017-11-26
**DEEP FUNCTION MACHINES: GENERALIZED NEURAL NETWORKS FOR TOPOLOGICAL LAYER EXPRESSION**

**Rating:** 7
**Confidence:** 1

**Review:**

This paper extends the framework of neural networks for finite-dimension to the case of infinite-dimension setting, called deep function machines. This theory seems to be interesting and might have further potential in applications.

---

> ### Author Response · Authors · 2018-01-04
> **Author Response**
>
> We would like to thank the reviewer for their time and useful summary.

---

### Official Review · AnonReviewer3 · 2017-11-27
**a functional analysis view of neural networks but without a major theorem**

**Rating:** 3
**Confidence:** 4

**Review:**

The main idea of this paper is to replace the feedforward summation
y = f(W*x + b)
where x,y,b are vectors, W is a matrix
by an integral
\y = f(\int W \x + \b)
where \x,\y,\b are functions, and W is a kernel. A deep neural network with this integral feedforward is called a deep function machine.

The motivation is along the lines of functional PCA: if the vector x was obtained by discretization of some function \x, then one encounters the curse of dimensionality as one obtains finer and finer discretization. The idea of functional PCA is to view \x as a function is some appropriate Hilbert space, and expands it in some appropriate basis. This way, finer discretization does not increase the dimension of \x (nor its approximation), but rather improves the resolution.

This paper takes this idea and applies it to deep neural networks. Unfortunately, beyond rather obvious approximation results, the paper does not get major mileage out of this idea. This approach amounts to a change of basis - and therefore the resolution invariance is not surprising. In the experiments, results of this method should be compared not against NNs trained on the data directly, but against NNs trained on dimension reduced version of the data (eg: first fixed number of PCA components). Unfortunately, this was not done. I suspect that in this case, the results would be very similar.

---

> ### Author Response · Authors · 2018-01-04
> **Author Response**
>
> We'd like to first express our gratitude for the reviewers time and useful comments. We will address the reviewers comments individually.
>
> >>> The main idea of this paper is to replace the feedforward summation [...] <<<
>
> While we introduce operator neural networks as such a formulation, the core idea of the paper is to unify the theory of infinite dimensional neural networks under one formulation, DFMs, and show that proving universal approximation results is ubiquitously reducible to the language of DFMs.
>
> >>> Unfortunately, beyond rather obvious approximation results, the paper does not get major mileage out of this idea. <<<
>
> We present DFMs with the explicit intention of providing a language for proving extremely difficult universal approximation results. While we agree that the statement of the results may be *simple*, as were the statements of the first universal approximation conjectures for discrete neural networks, our proofs resolve a major open-problem in the approximation theoretic literature which has gone unsolved for over 18 years now. (Stinchcombe, 1999), (Rossi et al., 2002), (Conan-Guez, Brieuc, 2002), etc.
>
> Our major theorem(s) are the reduction of these long-standing conjectures to the language of DFMs which provide a category theoretic framework for factorizing any universal approximation conjecture through that of the discrete DFM. (Theorem 3.5, Theorem 3.6, ***Definition 8.5, Lemma 8.6***).
>
>
> ### References ###
>
> Maxwell B Stinchcombe. Neural network approximation of continuous functionals and continuous
> functions on compactifications. Neural Networks, 12(3):467–477, 1999.
>
> Fabrice Rossi, Brieuc Conan-Guez, and François Fleuret. Theoretical properties of functional multi
> layer perceptrons. 2002.
>
> Christopher KI Williams. Computation with infinite neural networks. Neural Computation, 10(5):
> 1203–1216, 1998.
>
> Rossi, Fabrice, et al. "Representation of functional data in neural networks." Neurocomputing 64 (2005): 183-210.
>
> Conan-Guez, Brieuc, and Fabrice Rossi. "Approche régularisée du traitement de données fonctionnelles par un perceptron multi-couches." Actes des neuviemes journées de la SFC, Toulouse, France (2002): 169-172.

---

### Official Review · AnonReviewer1 · 2017-11-28
**This paper proposes a deep neural network approach to learn nonlinear operators. The current version of the paper requires significant improvements, both in terms of substance and in terms of presentation.**

**Rating:** 4
**Confidence:** 3

**Review:**

This paper deals with the problem of learning nonlinear operators using deep learning. Specifically, the authors propose to extend deep neural networks to the case where hidden layers can be infinite-dimensional. They give results on the quality of the approximation using these operator networks, and show how to build neural network layers that are able to take into account topological information from data. Experiments on MNIST using the proposed deep function machines (DFM) are provided.

The paper attempts to make progress in the region between deep learning and functional data analysis (FDA). This is interesting. Unfortunately, the paper requires significant improvements, both in terms of substance and in terms of presentation. My main concerns are the following:

1) One motivation of DFM is that in many applications data is a discretization of a continuous process and then can be represented by a function. FDA is the research field that formulated the ideas about the statistical data analysis of data samples consisting of continuous functions, where each function is viewed as one sample element. This paper fails to consider properly the work in its FDA context. Operator learning has been already studied in FDA. See for e.g. the problem of functional regression with functional responses. Indeed the functional model considered in the linear case is very similar to Eq. 2.5 or Eq. 3.2. Moreover, extension to nonparametric/nonlinear situations were also studied. The authors should add more information about previous work on this topic so that their results can be understood with respect to previous studies.

2) The computational aspects of DFM are not clear in the paper. From a practical computational perspective, the algorithm will be implemented on a machine which processes on finite representations of data. The paper does not clearly provide information about how the functional nature and the infinite dimensional can be handled in practice. In FDA, generally this is achieved via basis function approximations.

3) Some parts of the paper are hard to read. Sections 3 and 4 are not easy to understand. Maybe adding a section about the notation and developing more the intuition will improve the reading of the manuscript.

4) The experimental section can be significantly improved. It will be interesting to compare more DFM with its discrete counterpart. Also, other FDA approaches for operator learning should be discussed and compared to the proposed approach.

---

> ### Author Response · Authors · 2018-01-04
> **Reviewer Response**
>
> We thank the reviewer for their careful consideration and useful comments. We'd like to address a few points raised.
>
> 1) We agree that our treatment of existing empirical techniques relating to FDA and operator learning is minimal. However, the paper is presented as such so as to highlight the importance of the solution of the given universal approximation theorems in the language of DFMs and also unify existing infinite dimensional frameworks of neurocomputation. The appropriate references to existing linear/non-linear FDA works will be added in the camera-ready version.
>
> 2) Ultimately, any functional analytic approach to machine learning must have a computationally efficient discrete representation. In section 4.2 and Appendix D (10) we make substantial amount of time deriving those discrete representations. The code is currently released on Github but to preserve anonymity we will add a link after the review period has ended. We would like to highlight that again the primary purpose of the paper is in developing a new theoretical framework, DFMs, to show long standing unproven universal approximation results. The experimental and differential topological results are expositionally interesting corollaries of the structure of that language.
>
> 4) We would like to note that the experimental section is dedicated directly to comparing infinite dimensional DFMs with their discrete counterparts; that is, experiment 1 and experiment 2 compare to state of the art discrete neural networks respectively.

---

### Decision · Program_Chairs · 2018-01-29
**ICLR 2018 Conference Acceptance Decision**

**Decision:**

Reject

**Comment:**

The idea of extending deep nets to infinite dimensional inputs is interesting but, as the reviewers noted, the execution does not have the quality we can expect from an ICLR publication. I encourage the authors to consider the meaningful comments that were made and modify the paper accordingly.